# A Novel Approach for Detecting Unique Variations among Infectious Bacterial Species in Endocarditic Cardiac Valve Vegetation

**DOI:** 10.3390/cells9081899

**Published:** 2020-08-13

**Authors:** Nadji Hannachi, Hubert Lepidi, Anthony Fontanini, Tatsuki Takakura, Jacques Bou-Khalil, Frédérique Gouriet, Gilbert Habib, Didier Raoult, Laurence Camoin-Jau, Jean-Pierre Baudoin

**Affiliations:** 1Aix Marseille Univ, IRD, APHM, MEPHI, IHU Méditerranée Infection, 13005 Marseille, France; n_adji07@live.fr (N.H.); fontanini.anthony@gmail.com (A.F.); boukhaliljacques@gmail.com (J.B.-K.); Frederique.GOURIET@ap-hm.fr (F.G.); didier.raoult@gmail.com (D.R.); Laurence.CAMOIN@ap-hm.fr (L.C.-J.); 2Laboratoire D’anatomie et de Cytologie Pathologique, Hôpital de la Timone, AP-HM, boulevard Jean-Moulin, 13005 Marseille, France; Hubert.LEPIDI@ap-hm.fr; 3Hitachi High-Tech Corporation, Analytical & Medical Solution Business Group 882 Ichige, Hitachinaka-shi, Ibaraki-ken 312-8504, Japan; tatsuki.takakura.yk@hitachi-hightech.com; 4Département de Cardiologie, Hôpital de la Timone, AP-HM, Boulevard Jean-Moulin, 13005 Marseille, France; gilbert.habib3@gmail.com; 5Laboratoire D’Hématologie, Hôpital de la Timone, APHM, Boulevard Jean- Moulin, 13005 Marseille, France

**Keywords:** SEM, EDX, infectious endocarditis, bacteria, vegetation

## Abstract

Infectious endocarditis (IE) remains one of the deadliest heart diseases with a high death rate, generally following thrombo-embolic events. Today, therapy is based on surgery and antibiotic therapy. When thromboembolic complications in IE patients persist, this is often due to our lack of knowledge regarding the pathophysiological development and organization of cells in the vegetation, most notably the primordial role of platelets and further triggered hemostasis, which is related to the diversity of infectious microorganisms involved. Our objective was to study the organization of IE vegetations due to different bacteria species in order to understand the related pathophysiological mechanism of vegetation development. We present an approach for ultrastructural analysis of whole-infected heart valve tissue based on scanning electron microscopy and energy-dispersive X-ray spectroscopy. Our approach allowed us to detect differences in cell organization between the analyzed vegetations and revealed a distinct chemical feature in *viridans Streptococci* ones. Our results illustrate the benefits that such an approach may bring for guiding therapy, considering the germ involved for each IE patient.

## 1. Introduction

While important progress has been made in the diagnosis and treatment of cardiovascular diseases, infectious endocarditis (IE) remains one of the deadliest heart diseases with a high death rate [1]. It is an infection of the endocardium and heart valves or prosthetic valve implants caused mainly by cocci gram positive bacteria [2]. In short, our knowledge of the pathogenesis of this disease remains limited to the formation of a fibrin-platelet matrix that serves as a platform to receive the pathogen, followed by valve inflammation, all of which constitutes endocarditic vegetation [3]. Only a few studies that are available in the literature have shown interest in the mechanism of the development of the vegetation.

As platelets and fibrin play a key role in the formation of vegetation during IE, some researchers have investigated the use of anticoagulant agents for the treatment of IE to prevent the formation of fibrin [4,5]. Other studies have proposed the introduction of antiplatelet agents in order to inhibit platelet activation [6,7]. If the results of these studies were generally discordant and/or inconclusive, this might indicate that differences in the development process of IE are specific to the patient and the pathogen involved. In fact, we now know that bacterial species act with blood cells and particularly with platelets, in a different way from one another [8,9]. Thus, we hypothesized that the development of vegetation during IE may also depend on the bacterial species involved and the comorbid conditions specific to each patient.

Pathological examination of cardiac valves remains the gold standard for the diagnosis of IE [10]. Blood and valve cultures enable the precise identification of the pathogen involved [11]. In this study, we present a novel approach that consists of using a scanning electron microscopy (SEM) for the ultrastructural analysis of a control non-infected valve and five IE vegetations that involve different bacterial pathogens. We also used energy-dispersive X-ray spectroscopy (EDX) to examine the differences in the elemental composition of the vegetations. SEM-EDX is a powerful tool for biological research and diagnostics [12,13]. Recently, there was a rebirth of this technique in the field of biology, which enables the correlation of the ultrastructure and the chemical composition of cellular compartments in ultra-thin sections [14]. The objectives of this study were: (1) to rapidly detect the presence of bacteria in whole vegetation; (2) to understand the differences in the organization of the bacteria and the related inflammatory and hemostatic process within each vegetation: and (3) to detect possible differences in their chemical composition, depending on the bacterial species involved.

## 2. Material and Methods

### 2.1. Patients and Histological Analysis

Our analysis was performed on specimens obtained from six patients after cardiac surgery. One patient was a negative control with a degenerative valve without IE, presenting with severe mitral regurgitation related to mitral prolapse with flail leaflets, and five other patients diagnosed for IE, based on modified Duke criteria and due to one different and unique bacterial species (Table 1) [1]. Classical histological analysis was carried out during the diagnosis, using Giemsa, Grocott, Gram, periodic acid Schiff and Warthin–Starry staining.

### 2.2. Vegetation Preparation

Whole-cardiac valves pieces were fixed with glutaraldehyde 2.5% in 0.1 M sodium cacodylate buffer for at least 1 h. In order to access the depth of the tissue, vegetations were cut transversally with a razor blade in sterile conditions and then they were rinsed with 0.1 M sodium cacodylate buffer for 10 min and with distilled H_2_O for 10 min. The vegetations were gradually dehydrated with: ethanol 20% for 20 min; ethanol 50% for 10 min; ethanol 70% for 10 min; ethanol 85% for 20 min; ethanol 95% for 20 min; ethanol 100% for 30 min. Then the vegetations were incubated with ethanol 100%/hexamethyldisilazahne 100% in 1:2 ratio for 30 min (HDMS; Sigma-Aldrich; Ref. 379212) and with HDMS 100% for 10 min. A drop of HDMS 100% was dropped on top of the specimen and the vegetation was air-dried while protected from dust in a microbiological safety cabinet overnight. All solutions were 0.2 µm filtrated before use. The vegetations were mounted with their transverse cut facing upward on double-sided tape on clean glass slides and copper-coated with a MC1000 sputter coater (Hitachi) for 40 s at 10 mA.

### 2.3. Scanning Electron Microscopy and Energy-Dispersive X-ray Spectroscopy

Observation by SEM was either performed in a SU5000 scanning electron microscope (Hitachi High-Technologies, Tokyo, Japan) in low-vacuum at 15 keV with a BSE detector and observation mode (spot size 30) or a TM4000Plus table-top SEM (Hitachi High-Technologies, Tokyo, Japan) operated at 15 keV, lens mode 4 with a BSE detector. Magnification ranged between ×90 and ×11 000. A mean number of 173 images for each vegetation and 68 images for the degenerative valve were acquired. For semi-quantitative analysis (see results in Table 2), the relative abundance of each component (cocci, platelets, erythrocytes, leukocytes or fibrin) was determined by estimating the ratio of its presence with regard to the total number of images acquired for each valve, as follow: (−) *Absence:* no image found; (+) *Rare*: less than 10% of images; (++) *Moderate*: between 10% and 80% of images; (+++) *Abundant*: more than 80% of images.

EDX was performed with an INCA X-Stream-2 detector (Oxford Instruments) attached to the TM4000 SEM and AztecOne software (Oxford instruments) at 10mm working distance. Blind chemical mapping (all elements considered) on random regions of the valves were performed in the first round of observations, and selected elements (carbon, nitrogen, oxygen, calcium and phosphorus) were mapped in the second time on selected regions (bright versus non-bright regions). Weight and atomic percentages were retained for quantitative study.

### 2.4. Statistical Analysis

GraphPad (Prism) for Windows was used for statistical analysis of the EDX data. As samples were normally distributed, significant differences were determined using the two-tailed, unpaired student’s *t* test. Statistical significance was set at *p* < 0.05.

## 3. Results

### 3.1. Histological and Immunohistochemical Observations

Six valves were considered for analysis. One valve served as a negative control provided from a patient without IE, who had severe mitral regurgitation related to mitral prolapse with flail leaflets. The five other valves corresponded to IE cases with vegetation. Out of these five vegetations, histological analysis reported an inflammatory state typical of infectious endocarditis for three of them (*Enterococcus faecalis*, *Staphylococcus aureus* and *Streptococcus oralis*). A discrete inflammation was reported for *Streptococcus agalactiae* and *Streptococcus gallolyticus*. The presence of bacteria was reported for *S. oralis* and *S. aureus*, but not for *E. faecalis*, *S. agalactiae* and *S. gallolyticus*. Blood culture was used to identify the bacterial species involved in each IE case [15].

### 3.2. Scanning Electron Microscopy

We used SEM to understand the cellular organization on top and inside the analyzed vegetations. We found various tissue and cellular organizations depending on the bacterial species involved by comparing the amount of platelets, fibrin, bacteria and leukocytes, and the respective cellular arrangements of the different elements within the vegetation (Table 2).

### 3.3. Presence and Organization of Bacteria in the Vegetative Tissue

First, we made sure of the absence of bacteria in the degenerative valve serving as a negative control (Appendix A). Second, we looked for the presence of bacteria in the endocarditic vegetations. Bacteria appeared as round objects, isolated or arranged in clusters. The mean maximal diameter of bacteria ranged between 750 to 800 nm, which is consistent with their morphological characteristics [9]. In contrast to the cases of *E. faecalis,* where the bacteria were present in low amounts (less than 10% of our images) (Figure 1), *S. aureus*, *S. oralis*, *S. agalactiae* and *S. gallolyticus* (Figure 2, Figure 3, Figure 4 and Figure 5) were present in larger numbers in the other vegetations. We also noticed a significant number of dividing bacteria in the case of *S. oralis*, where the bacteria presented typical proliferation features such as double cell bodies and division septa (Figure 4B). Because the SEM screening of large regions of the surface and the depth of the IE cardiac vegetations comes directly from the operative bloc, we were able to detect bacteria in all the five IE vegetations analyzed. Even if the histological analysis had been negative for finding such objects in the cases of *E. faecalis*, *S. agalactiae* and *S. gallolyticus*, SEM proved to be efficient in finding infectious bacteria in all five vegetations.

### 3.4. The Different Extent of Inflammation

The extent of the inflammation was estimated by the abundance of leukocytes, which likely corresponded, based on morphology, to polymorphonuclear neutrophils and macrophages. These cells were identified by their globular shape and diameters ranging from 12 to 20 µm. The inflammatory state was more pronounced in the case of *E. faecalis*, *S. agalactiae* and *S. aureus*, (between 10% and 80% of analyzed images in the case of *E. faecalis* and *S. agalactiae* and more than 80% in the case of *S. aureus* vegetation) (Figure 1, Figure 2 and Figure 3) compared to the other vegetations. The inflammatory cells mostly appeared as individualized or embedded in the tissue. They were located in regions that also contained bacteria in the case of *S. aureus* and *S. agalactiae* (Figure 2D,F and Figure 3D)*,* and *E. faecalis*, despite a lower level of bacteria in the latter case. In the vegetations with *S. gallolyticus* and *S. oralis*, a few inflammatory cells were observed, and these were mainly isolated.

### 3.5. Platelet Recruitment and Extent of the Fibrin Network

Firstly, the overall appearance of the vegetation was mainly amorphous and filamentous in the case of the *Streptococcus* and *S. aureus* species. The filaments had the appearance of a fishing net in the latter case (Figure 2C). The vegetation with *E. faecalis* as well as with the degenerative valve had a stratified appearance (Appendix A and Figure 1) and was devoid of filaments. Platelets appeared with a diameter between 1.5 and 3 µm, with or without protrusions. They were observed in large numbers with the *Streptococci* and *S. aureus* vegetation, (between 10% and 80% of analyzed images in the case of *S. agalactiae* and more than 80% in the case of *S. aureus, S. oralis* and *S. gallolitycus* vegetation). In these vegetations, platelets were either individual, caught in the fibrin network, or rather aggregated in clusters, with loss of cellular integrity and an amorphous platelet matrix (Figure 2, Figure 3, Figure 4 and Figure 5). The platelet/bacteria ratio was approximately 1:1 in all vegetations, except in the case of *S. oralis*, where this ratio was approximately one platelet per 10 bacteria. We noticed that in the case of *E. faecalis*, erythrocytes were much more abundant than platelets (Figure 1). Erythrocytes were recognized based on their mean diameter of 7 µm and their characteristic biconcave donut shape.

There seemed to be a correlation between the abundance of the fibrin network and the number of platelets in the different vegetations. Indeed, large networks of fibrin were observed in *S. aureus* (Figure 2), *S. agalactiae* (Figure 3) and *S. oralis* (Figure 4D,E) vegetation, surrounding both bacteria and platelets, and also erythrocytes in the case of *S. agalactiae* and *S. aureus* vegetation. Fibrin filaments were less abundant in the case of *S. gallolyticus* vegetation. Regarding *E. faecalis* vegetation, the fibrinous networks were very reduced and limited only to a few narrow regions (Figure 1D).

Finally, we also noticed regions with a remarkably bright appearance when imaging the *S. oralis* and *S. gallolyticus* vegetation (Figure 4A, Figure 5C and Appendix A). When zooming-in, these bright regions under the SEM electron beam had well defined outlines and were mainly composed of platelets and bacteria. In some of these regions, cocci-like ghost shapes were detected.

### 3.6. Energy Dispersive X-Ray Spectroscopy

To decipher the nature of the bright regions observed in the *S. oralis* and *S. gallolyticus* vegetation, we proceeded to EDX analysis. Surprisingly, EDX revealed that the bright regions emitted strong signals for calcium (Ca) and phosphorus (P) compared to the other regions (Figure 6A). This resulted in significant peaks for these two elements in the EDX spectra (Figure 6B). As shown in Table 3, which provides the values obtained following the analysis of the *S. gallolyticus* vegetation, significant changes in the weight and atomic percentage of C, O, Ca and P were observed between non-bright (*n* = 5 for each element) and bright regions (*n* = 9 for each element). Indeed, the measurements carried out on the bright regions showed a decrease in the abundance of carbon and an increase in the abundance of oxygen, phosphorus and calcium. There was no difference in nitrogen between the two types of region (Table 3). Analysis of the other vegetation as well as the degenerative valve using the EDX tool showed a low signal emission with regard to calcium and phosphorus (Appendix A).

To better understand the possible nature of the compound(s) responsible for such an increase in Ca and P in these regions, we calculated the average weight and atomic Ca/P ratios in the bright regions of interest in *S. gallolyticus* vegetation (*n* = 9). We found a mean Ca/P weight percentage ratio of 2.18 ± 0.29 and a mean Ca/P atomic percentage ratio of 1.52 ± 0.34, both values that are close to those calculated for hydroxyapatite compounds in biological samples [16]. Similar values were obtained from the *S. oralis* vegetation analysis (Appendix A).

## 4. Discussion

Our study reported different cell organization between the analyzed endocardial vegetations (Table 2). These differences involved, on the one hand, the hemostatic state represented by the abundance of platelets and the extension of the fibrin network, and on the other hand, the extension of the inflammatory state represented by the abundance of leukocytes.

First, the influence of the bacterial species on the ultrastructure of platelet aggregates has been previously demonstrated in vitro, by light and electron microscopy [9]. *E. faecalis* induced loose aggregates, with platelets that fully retained their cellular integrity while *S. sanguinis* and *S. aureus* were associated with very dense aggregates and coalesced platelets. Accordingly, we showed here that the vegetation induced by *E. faecalis* was formed by a few platelets and fibrin, which suggested low engagement in the hemostatic process, compared to the vegetation due to *S. aureus* and streptococci species, which were associated with many more platelets and fibrin.

Second, we showed that the inflammatory state of the vegetations was different, with very inflammatory vegetations, such as the ones associated with *S. aureus*, *E. faecalis* and *S. agalactiae* and less inflamed vegetations associated with *S. gallolyticus* and *S. oralis*. Accordingly, group B streptococci (*S. agalactiae*) have been shown to activate neutrophils [17]. In another comparative study, it was reported that the activation of neutrophils induced by *S. aureus* and *E. faecalis* was relatively greater than that induced by viridans streptococci [18].

Third, EDX analysis showed some unexpected results regarding the chemical composition of endocarditic vegetation. Viridans streptococci vegetations (*S. gallolyticus* and *S. oralis*) were associated with a bright deposit, composed mainly of Ca and P, with a ratio close to that of hydroxyapatite. Such calcified deposits in cardiac valves have been previously reported [19], especially in the vegetation due to *S. salivarius*, which is also part of viridans streptococci [20]. Normally, Ca-P deposits occur in the bones and teeth, but also they can be found abnormally in other soft tissues, and heart and urinary stones [21,22]. Interestingly, in such stones, bacteria ghosts were found, such as the ones we imaged in our samples [23]. Although calculogenic substances in the body fluids and/or lipids may be causally linked to the Ca-P deposits we observed [21,24], the presence of bacteria proliferation features and of bacteria ghosts inside the Ca-P deposits tends to link the bacteria to the development of these deposits in the case of IE vegetation. We hypothesize that the development of Ca-P deposits may primarily be an infectious bacteria species-dependent phenomenon, due to the specific activity of the viridans streptococci bacteria. It may ultimately influence the consistency of the vegetation and may have a role in further appearance of emboli.

In conclusion, the use of SEM on whole tissues made the detection of bacteria relatively fast with no need for special staining or time-consuming tissue embedding/ultramicrotomy methods. EDX provides a chemical signature for the tissues infected with specific species, which eventually will help to define new strategies for rapid diagnosis of the bacterial infection in the heart.

We are aware that the analysis of a single vegetation per bacterial species cannot establish a firm relationship between ultrastructure-bacteria, something that would be interesting to study further. Nonetheless, through this study we are able to highlight that the composition of endocardial vegetation is far from being universal, with infected and inflamed fibrin-platelet deposit, but rather needs to be analyzed considering both the bacterial species involved and the characteristics of the patient. Indeed, we suggest a divergent pathogenesis for IE, which could act as a guide for personalized therapy depending on the infectious agent involved. Meanwhile, the patient’s history may also influence the vegetation development/organization, for example, pre-operative antibiotics and anti-thrombotic treatments. Also, here we worked with vegetations recovered from mitral valves and aortic valves, and this may also be a factor that differentially influences the development of the vegetation.

## Figures and Tables

**Figure 1 cells-09-01899-f001:**
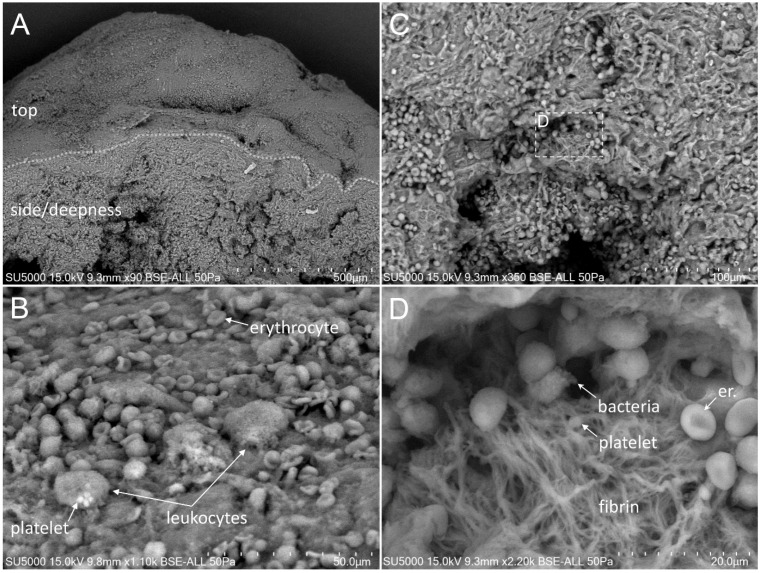
Scanning electron microscopy (SEM) of *E. faecalis*-infected vegetation. (**A**) Low magnification showing the top and depth of the vegetation. (**B**) Surface view of the vegetation, with numerous erythrocytes and a few platelets and leukocytes. (**C**) Low magnification view of the vegetation side depth, showing the stratified organization of intermingled erythrocytes inside a network of fibrin bundles. (**D**) Zoom-in on the boxed region in (**C**), with erythrocytes, a cluster of bacteria and some platelets mixed with the fibrin network.

**Figure 2 cells-09-01899-f002:**
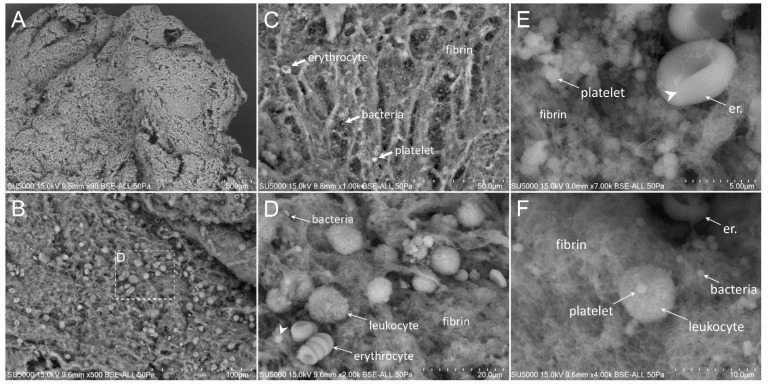
Scanning electron microscopy of *S. aureus*-infected vegetation. (**A**,**B**) Low magnification of the vegetation. The vegetation had a filamentous appearance with numerous leukocytes. (**C**) The vegetation is rich in platelets and very thin fibrin fibers, giving the appearance of a fishing net (**D**) Zoom-in on the boxed region in (**B**) showing leukocytes, bacteria and platelets (arrowhead) in a thin fibrinous network. (**E**) Aggregated platelets next to an erythrocyte enwrapped with fibrin fibers (arrowhead). (**F**) Platelets on top of a leukocyte.

**Figure 3 cells-09-01899-f003:**
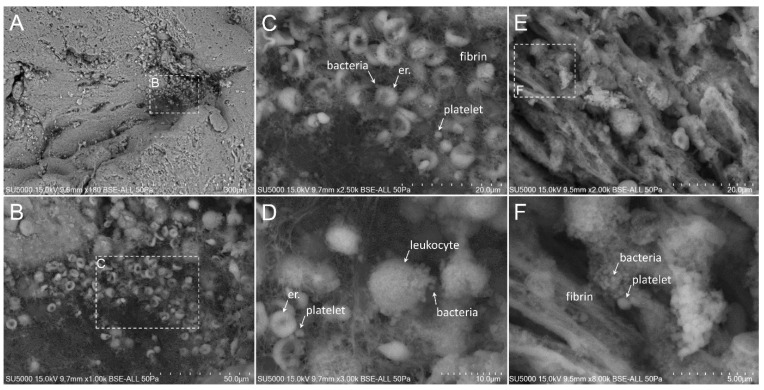
Scanning electron microscopy of *S. agalctiae*-infected vegetation. (**A**,**B**) Low magnification of the vegetation showing the presence of a high number of erythrocytes. (**C**) Erythrocytes, platelets and bacteria enwrapped in a thin fibrin network. (**D**) Leukocytes lined by bacteria. (**E**) Thick fibrin bundles with aggregates of platelets and clustered bacteria. (**F**) Zoom-in on the boxed region in (**E**).

**Figure 4 cells-09-01899-f004:**
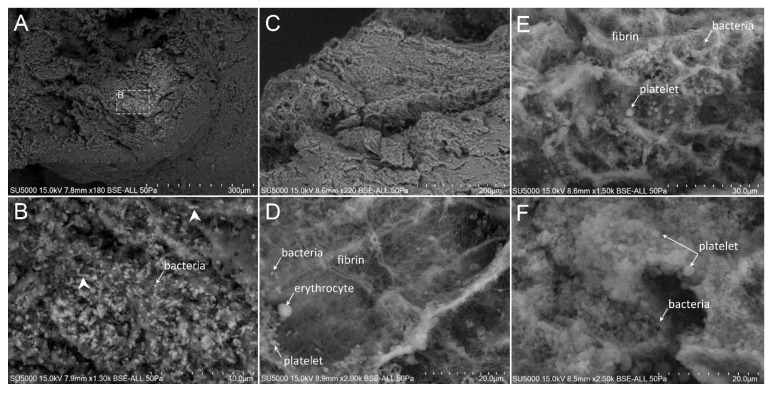
Scanning electron microscopy of *S. oralis*-infected vegetation. (**A**) Low magnification centered on a bright region of the vegetation. (**B**) Zoom-in on the boxed region in (**A**), showing numerous bacteria with typical proliferation features (arrowheads). (**C**) Low magnification of another region of the vegetation. (**D**) High magnification of the same region in (**C**) showing the fibrin network and its thin filaments or thicker bundles. (**E**) High magnification of platelets and bacteria mixed with fibrin. (**F**). Bacteria and platelet aggregates next to fibrin-rich regions.

**Figure 5 cells-09-01899-f005:**
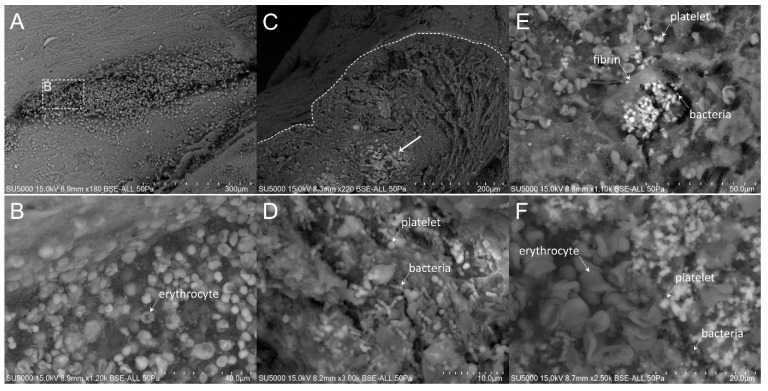
Scanning electron microscopy of *S. gallolyticus*-infected vegetation. (**A**) Low magnification of the surface of the vegetation. (**B**) Zoom-in on the boxed region in (**A**), with erythrocytes more or less visible in an amorphous cloth. (**C**) Surface and depth of the vegetation, which presented a layered organization at its periphery and an internal bright region (arrow). (**D**) High magnification on the bright region in (**C**) showing platelets and bacteria cocci chains. (**E**) High magnification showing platelets and bacteria next to a bundle of fibrin. (**F**) Damaged erythrocytes, platelets and bacteria. The vegetation is rich in platelets but with a sparse fibrinous network. Most of the cells were found embedded in an amorphous cloth.

**Figure 6 cells-09-01899-f006:**
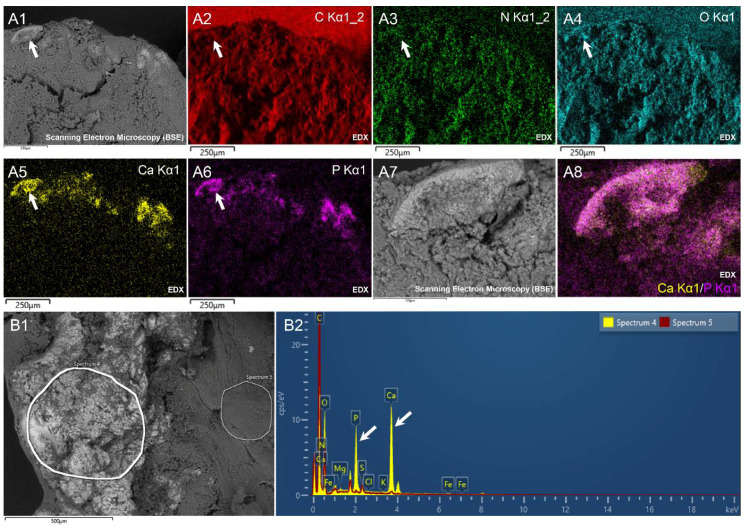
Scanning electron microscopy coupled to energy-dispersive X-ray (EDX) spectroscopy of a bright region from *S. oralis* and *S. gallolyticus* infected vegetation. (**A**) *S. oralis* vegetation with bright regions. (**A1**) SEM view. (**A2**–**A6**) Multi-element EDX mapping images of C, N, O, Ca and P. (**A7**) High magnification view by SEM of the region depicted in (**A1**–**A6**) (arrow). (**A8**) Colocalization of Ca and P-EDX map of the same region as in (**A7**). (**B**) *S. gallolyticus* vegetation. (**B1**) SEM image with large bright regions. (**B2**) EDX spectra of the two regions circled in (**B1**), showing the higher amount of P and Ca for spectrum 4 compared to spectrum 5.

**Table 1 cells-09-01899-t001:** Demographic and clinical data at admission.

	Patient 1	Patient 2	Patient 3	Patient 4	Patient 5	Patient 6
Clinical data						
**Sex**	Male	Male	Male	Female	Male	Female
**Age (Y)**	88	87	52	93	71	67
**Weight (kg)**	78	84	83	52	75	64
**Diabetes**	No	Yes	No	No	No	No
**Smoking**	No	No	Yes	No	Yes	No
**Hemodialysis**	No	No	No	No	No	No
**Myocardial infarct**	No	No	No	No	No	No
**History of stroke**	No	No	No	No	No	No
**HTA**	Yes	Yes	No	Yes	Yes	No
**Atrial fibrillation**		Yes	No	No	No	No
**Hist. of malignancy**	No	No	No	No	No	Yes
**IV drug use**	No	No	No	No	No	No
Localization	Mitral valve	Aortic valve	Mitral valve	Mitral valve	Mitral valve	Aortic valve
Echocardiography						
**Vegetation**	No	Aortic	Mitral	Mitral	Mitral	Aortic
**Vegetation length (mm)**	N/A	6	30	20	18	13
**Severe regurgitation**	Yes	No	No	No	No	No
**Valve perforation**	No	No	No	No	Yes	Yes
**Abscess**	No	Yes	No	No	Yes	No
Antibiotherapy	No	Amoxicilline cefriaxone	Daptomycine Sulfamétroxazole trimetroprime	ceftriaxone	Gentamycine and ceftriaxone	Gentamycine and amoxicilline
**Duration of antibiotherapy before surgery (days)**	No	9 days	No	15 days	15 days	15 days
**Biology**						
**Hemoglobin (g/L)**	14.7	116	105	120	99	111
**Platelets (G/L)**	161	214	173	198	244	230
**CRP (mg/L)**	10	180	118	118	52	98
**Creatinine (** **μmol/L)**	166	67	81	81	76	94.5
**BNP (ng/L)**	900	405	207	328	534	1336
**Ca (mmol/L)**	2.10	2.05	2.01	1.99	2,19	2.04
**P (mmol/L)**	1.5	1.34	0.9	1.08	1.02	1.34
Bacteriology	Negative	*E. faecalis*	*S. aureus*	*S. agalactiae*	*S. Gallolyticus*	*S. oralis*

Patients were considered to have confirmed infective endocarditis if microorganisms were detected by standard blood cultures or cultures of valvular material, according to the application of the Duke criteria [1].

**Table 2 cells-09-01899-t002:** Differential cellular composition of the endocarditic vegetations according to the bacterial species involved.

Species	General Aspect	Cocci	Platelets	Fibrin	Erythrocytes	Leukocytes	Platelets/Bacteria Ratio
(Degenerative valve)	Fibrous laminate, few cells	−	−	−	++ (On the surface)	−	−
*E. faecalis*	Amorphous laminate	+	+	+	+++ (Major component)	++	1
*S. aureus*	Amorphous and filamentous	+++	+++	+++	++ (Incorporated into the fibrin network)	+++	1
*S. agalactiae*	Amorphous and filamentous	++	++	+++	++ (Incorporated into the fibrin network)	++	1
*S. gallolyticus*	Amorphous and filamentous	+++	+++	+	++ (On the surface)	+	1
*S. oralis*	Amorphous and filamentous	+++	+++	+++	++ (On the surface)	+	10

(−): Absence. (+): Rare. (++): Moderate. (+++): Abundant.

**Table 3 cells-09-01899-t003:** Results (mean ± SD) of weight and atomic percentages of C, O, N, P, Ca in the bright and non-bright regions of the *S. gallolyticus* vegetation. ** *p* < 0.01; *** *p* < 0.001.

		C	O	N	Ca	P
Weight %	Non-bright regions	**60.2** ± 2.05	**21.4** ± 0.64	**17.75** ± 2.4	**0.15** ± 0.12	**0.08** ± 0.06
	Bright regions	**49.95** ± 3.28 ***	**29.88** ± 3.72 ***	**15.88** ± 1.46	**2.63** ± 0.61 ***	**1.22** ± 0.28 ***
Atomic %	Non-bright regions	**65.62** ± 2.05	**17.52** ± 0.55	**16.6** ± 2.27	**0.05** ± 0.04	**0.03** ± 0.026
	Bright regions	**57.09** ± 3.05 ***	**24.85** ± 4.72 **	**16.4** ± 3.09	**0.9** ± 0.22 ***	**0.65** ± 0.36 **

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
