# Peer review of "A Novel Approach for Detecting Unique Variations among Infectious Bacterial Species in Endocarditic Cardiac Valve Vegetation"

_cells, 2020, doi:10.3390/cells9081899_

Round 1
Reviewer 1 Report
Manuscript ID: cells-869002
The manuscript entitled “Ultrastructural analysis of endocarditic cardiac valve vegetations reveal unique variations among infectious bacterial species” by Hannachi, et al., aims to determine the structural composition of human cardiac valve vegetations upon infectious endocarditis (IE) due to different bacterial species. IE is life-threatening and the development of its associated bacterial vegetation is not well understood mechanistically. Based on previous studies showing IE vegetations are comprised of platelets and fibrin and that different bacterial species differentially affect platelet activation, the authors hypothesize that vegetation development and ultrastructure in IE is dependent on the species of the infecting bacteria. Using scanning electron microscopy and energy-dispersive x-ray spectrometry, the data provided suggests there is a correlation in the cellular and atomic composition of valve vegetation and the co-infecting bacterial species. These findings are interesting and could provide important insights into bacteria-specific causes underlying IE. However, the data as presented are limited and observational in its current form and could be significantly improved by addressing the following limitations.
Major Issues:
- Conclusions appear to be based on n=1 valve per bacterial species versus an n=1 control degenerative valve. Without analyzing valves from multiple patients with IE due to the same bacterial species, this study is strictly observational and conclusions regarding any structural, cellular, and chemical vegetation differences cannot be made.
- Secondary validation of IE patient presentation and histological valve structure are not provided. Section 2.1 in the Materials and Methods state diagnosis by modified Dukes criteria and classical histological analysis was performed (but are not given). These data are important for data interpretation, associate with disease severity and to independently verify the structures and cell types identified by electron microscopy.
- Whether the same or different cardiac valve(s) from different hearts are investigated is not specified.
- EDX findings examining oralis and S. gallolyticus show these two viridian streptococci vegetations have high Ca and P abundance. It is concluded that Ca-P deposits may develop primarily in a species dependent manner. Without quantifying EDX spectra in IE tissues from other species and from control tissues, such conclusions cannot be drawn.
Minor Issues:
- It is assumed, but not explicitly stated, that each of human patients with IE was infected with a different unique bacterial species. Is this correct?
- SEM findings in sections 3.3-3.5 are described using subjective terms (e.g. more, less, higher, lower, etc.). It appears quantitative data are available and used to generate Table 1. It would be informative to provide such numbers in these sections.
- There is no arrow in panel C of Figure 5 as stated in the legend.
- The clinical implications of finding different structures, cell types, and atomic abundances in IE tissues due to different bacterial species are not discussed. Are there differences in clinical presentation, treatment regimens or outcomes in IE that are dependent on the infecting species? Are there differences in the response to anti-thrombotic or anti-platelet therapies?
- The Introduction could benefit from expanding on IE and defining some of the more technical terms to make it more readable for a broader audience.
Author Response
Comments and Suggestions for Authors
Manuscript ID: cells-869002
The manuscript entitled “Ultrastructural analysis of endocarditic cardiac valve vegetations reveal unique variations among infectious bacterial species” by Hannachi, et al., aims to determine the structural composition of human cardiac valve vegetations upon infectious endocarditis (IE) due to different bacterial species. IE is life-threatening and the development of its associated bacterial vegetation is not well understood mechanistically. Based on previous studies showing IE vegetations are comprised of platelets and fibrin and that different bacterial species differentially affect platelet activation, the authors hypothesize that vegetation development and ultrastructure in IE is dependent on the species of the infecting bacteria. Using scanning electron microscopy and energy-dispersive x-ray spectrometry, the data provided suggests there is a correlation in the cellular and atomic composition of valve vegetation and the co-infecting bacterial species. These findings are interesting and could provide important insights into bacteria-specific causes underlying IE. However, the data as presented are limited and observational in its current form and could be significantly improved by addressing the following limitations.
Dear reviewer, we agree with your summary and we hope to address all your concerns below.
Major Issues:
- Conclusions appear to be based on n=1 valve per bacterial species versus an n=1 control degenerative valve. Without analyzing valves from multiple patients with IE due to the same bacterial species, this study is strictly observational and conclusions regarding any structural, cellular, and chemical vegetation differences cannot be made.
Response 1:
We fully agree. The primary message of this article is that of the presence of several possible cellular organizations within endocarditic vegetations. In clinical studies previously carried out on the potential usage of antithrombotics in infectious endocarditis, this pathology was considered as a single entity, no taking in consideration the species involved, which led to contradictory and inconclusive results. Through this article, we draw attention to the possibility of the presence of divergences in the pathophysiology of infective endocarditis and where the pathogen could be a determining factor. As you made the point, we remained cautious about the conclusions, something which must be confirmed by future studies.
From our point of view, one important point exposed in this article is the provision of a new analytical method allowing on the one hand to study the cellular and chemical organization of the vegetations and therefore the pathophysiology of infective endocarditis and on the other hand, it could be added to histology analysis in the stages of diagnosis.
Accordingly, we changed our manuscript title from “Ultrastructural analysis of endocarditic cardiac valve vegetations reveal unique variations among infectious bacterial species” to “A novel approach reveals unique variations among infectious bacterial species in endocarditic cardiac valve vegetations”. Indeed, this may help readers to consider our article as preliminary regarding results but establishing a promising novel approach. We also added in the introduction references to conventional methods for IE diagnosis.
- Secondary validation of IE patient presentation and histological valve structure are not provided. Section 2.1 in the Materials and Methods state diagnosis by modified Dukes criteria and classical histological analysis was performed (but are not given). These data are important for data interpretation, associate with disease severity and to independently verify the structures and cell types identified by electron microscopy.
Response 2:
We agree with this remark and we provided more details about vegetation localization. Please check the table 1 in the new submitted version. For the modified duck criteria, we directed towards a reference of our team detailing these diagnostic criteria. For histology, this represents the first part of the results.
- Whether the same or different cardiac valve(s) from different hearts are investigated is not specified.
Response 3:
The vegetations analyzed were associated with two different locations, the mitral or the aortic valves. We have clarified this point in Table 1 of the new version of the manuscript. Interestingly, the two vegetations associated with phospho-calcic deposits were also associated with different locations.
- EDX findings examining oralis and S. gallolyticus show these two viridian streptococci vegetations have high Ca and P abundance. It is concluded that Ca-P deposits may develop primarily in a species dependent manner. Without quantifying EDX spectra in IE tissues from other species and from control tissues, such conclusions cannot be drawn.
Response 4:
We have already analyzed the other vegetations and the degenerative valve using the EDX tool. The signals emitted from these samples regarding calcium and phosphorus were very weak. We provide here a new figure, added as supp-fig 3 showing these transmitted signals. Please check the new submitted version. Tiny white spots appear in the condition with S. agalactiae, giving signals of Ca and P. We consider these spots as debris deposited on the surface of the sample and we did not take them into account.
Measurement of weight and atomic percentages of C, Ca and P in one representative region from the control degenerative valve as well as from each of the other vegetations has showed week presence of Ca and P in these samples.
|
|
Weight% |
Atomic% |
||||
|
|
C |
Ca |
P |
C |
Ca |
P |
|
Degenerative valve |
56.67 |
0.02 |
0.02 |
62.22 |
0.01 |
0.01 |
|
E. faecalis |
59.98 |
0.02 |
0.05 |
65.76 |
0.01 |
0.02 |
|
S. aureus |
60.02 |
0.02 |
0.02 |
65.8 |
0.01 |
0.01 |
|
S. agalactiae |
57.53 |
0.05 |
0.06 |
63.02 |
0.02 |
0.03 |
Minor Issues:
- It is assumed, but not explicitly stated, that each of human patients with IE was infected with a different unique bacterial species. Is this correct?
Response 1:
Effectively, each patient was infected by a different unique bacterial species. We have specified this point in the new submitted version, line: 77-78; page: 5.
- SEM findings in sections 3.3-3.5 are described using subjective terms (e.g. more, less, higher, lower, etc.). It appears quantitative data are available and used to generate Table 1. It would be informative to provide such numbers in these sections.
Response 2:
Done as requested, please check these sections in the new submitted version. However, we think that we have to maintain these adjectives because we performed a comparative analysis.
- There is no arrow in panel C of Figure 5 as stated in the legend.
Response 3:
We apologize for this careless error. We corrected this point in the new submitted version.
- The clinical implications of finding different structures, cell types, and atomic abundances in IE tissues due to different bacterial species are not discussed. Are there differences in clinical presentation, treatment regimens or outcomes in IE that are dependent on the infecting species? Are there differences in the response to anti-thrombotic or anti-platelet therapies?
Response 4:
Indeed, this presents the major point of our article. Currently, infectious endocarditis is assumed to be a single pathophysiological entity, not considering the bacterial species involved, except in the choice of antibiotic therapy. It is until now assumed that infectious endocarditis is a set of complications triggered by the presence of fibrino-platelet-infected vegetation. To date, antithrombotic drugs are not used. We recently published articles demonstrating that antiplatelet agents act differently on the inhibition of platelet aggregation depending on the inducing bacterial species.
The strong message of this current article is to show a difference in the ultrastructure of the vegetations itself, which could be linked to the bacterial species involved, and which suggests divergent pathophysiology, giving the idea to be treated in a more precise and personalized manner.
- Hannachi N, Baudoin JP, Prasanth A, Habib G, Camoin-jau L. (2019a). The distinct effects of aspirin on platelet aggregation induced by infectious bacteria. 1-11. DOI: 10.1080/09537104.2019.1704717
- Hannachi N, Ogé-Ganaye E, Baudoin JP, Fontanini A, Bernot D, Habib G, Camoin-Jau Antiplatelet Agents Have a Distinct Efficacy on Platelet Aggregation Induced by Infectious Bacteria. Front Pharmacol. 2020; 11: 863. doi: 10.3389/fphar.2020.00863
- The Introduction could benefit from expanding on IE and defining some of the more technical terms to make it more readable for a broader audience.
Response 5:
Done as requested. Please check the new submitted version, lines: 59-68; pages: 3 and 4.

Reviewer 2 Report
In this manuscript, Hannachi et al. present a study of the complex biofilms formed by bacterial infections of the cardiac valves during infectious endocarditis. They employ SEM and EDX to demonstrate an ultrastructure analysis of vegetation development. Little is known about the mechanistic pathogenesis of valvular infectious endocarditis, though it is believed that different bacterial species interact with blood cells and platelets in different manners to produce the platelet/fibrin framework for vegetation growth in IE. The work described herein is novel and of substantial importance. The manuscript is clear, concise, and well-written. However, the lack of biological replicates and descriptive nature of the histological findings impact this manuscript’s suitability for publication.
Major concerns:
-A significant and substantial limitation is the absence of biological replicates in this work. Dramatic donor-to-donor variability in valvular morphology is known to be present in both normal and diseased valve leaflets. At a minimum, n=3 biological replicates per condition should be included. Furthermore, there is known association of altered valvular homeostatic mechanisms, ultrastructure, etc with age, sex, and other co-morbidities (e.g. lipid levels, kidney function, etc). At a minimum, these basic clinical data should be matched between control valves and IE valves, and/or disclosed in the manuscript to enable proper assessment of baseline characteristics.
-Further details of the “degenerative valve without IE” that acted as a control should be described – why was this valve removed? Many valvular pathologies (aortic regurgitation, aortic stenosis, etc) are associated with altered/elevated thrombosis, while other common sources of “non-calcified” or “non-fibrotic” valves (i.e. discarded hearts from patients that have undergone heart transplantation) are likely to be non-normal, due to altered cardiac hemodynamics, increased circulating calcium/phosphate due to concomitant kidney failure, etc.
-Were tissues extracted from donors and processed for histology in fully-aseptic conditions? How was the possibility of environmental contamination excluded? Especially in light of the relatively-low degree of bacteria found in several vegetations…
-Blood culture methodology used to confirm bacterial species found in IE valves is not described in the Materials and Methods. Were cultures taken from the valve leaflets themselves, from donor blood samples, etc?
-Manual quantitation of cellular organization, cell types, and their respective arrangements as imaged by SEM is basic and prone to bias. The hisological results presented in this manuscript are largely descriptive. With the widespread availability of automatic image analysis algorithms (inexpensive & open-source eg ImageJ/Fiji, or turnkey options from most microscope vendors), automatic quantitation of these parameters is desirable, as it enables quantifiable and blinded insight into these parameters.
-Were investigators blinded to valve type/infection type during semi-quantitative analyses of disease components by histology?
-Why were multiple student’s t tests utilized for comparisons of EDX data? This type of multiple-comparisons testing requires ANOVA or similar.
-Without further clinical patient data and n=1 biological replicates per condition, the purported presence of calcific deposits in veridans streptococci-infected valves is of little value. Calcium phosphate deposits are a common finding in diseased leaflets, and could easily be present due to underlying AI, AS, or other fibrocalcific valvular disease, kidney complications, etc.
Author Response
Comments and Suggestions for Authors
In this manuscript, Hannachi et al. present a study of the complex biofilms formed by bacterial infections of the cardiac valves during infectious endocarditis. They employ SEM and EDX to demonstrate an ultrastructure analysis of vegetation development. Little is known about the mechanistic pathogenesis of valvular infectious endocarditis, though it is believed that different bacterial species interact with blood cells and platelets in different manners to produce the platelet/fibrin framework for vegetation growth in IE. The work described herein is novel and of substantial importance. The manuscript is clear, concise, and well-written. However, the lack of biological replicates and descriptive nature of the histological findings impact this manuscript’s suitability for publication.
Dear reviewer, we agree with your summary and we hope to address all your concerns below.
Major concerns:
-A significant and substantial limitation is the absence of biological replicates in this work. Dramatic donor-to-donor variability in valvular morphology is known to be present in both normal and diseased valve leaflets. At a minimum, n=3 biological replicates per condition should be included. Furthermore, there is known association of altered valvular homeostatic mechanisms, ultrastructure, etc with age, sex, and other co-morbidities (e.g. lipid levels, kidney function, etc). At a minimum, these basic clinical data should be matched between control valves and IE valves, and/or disclosed in the manuscript to enable proper assessment of baseline characteristics.
Response 1:
Dear reviewer, we agree that having replicates for each species would strengthen our hypothesis. However, as you know, infective endocarditis presents a rare disease, with a multitude of species that can be involved. Therefore, having replicates of vegetation due to specific species, knowing that some are very rare, can be a real obstacle, and can significantly delay the exploitation of data.
From our point of view, one important point exposed in this article is the provision of a new analytical method allowing on the one hand to study the cellular and chemical organization of the vegetations and therefore the pathophysiology of infective endocarditis and on the other hand, it could be added to histology analysis in the stages of diagnosis.
Regarding information related to patients, age, sex, renal function ..., we are fully supportive, and we have included this in the new version submitted, please check the Table 1 in the new submitted version.
-Further details of the “degenerative valve without IE” that acted as a control should be described – why was this valve removed? Many valvular pathologies (aortic regurgitation, aortic stenosis, etc) are associated with altered/elevated thrombosis, while other common sources of “non-calcified” or “non-fibrotic” valves (i.e. discarded hearts from patients that have undergone heart transplantation) are likely to be non-normal, due to altered cardiac hemodynamics, increased circulating calcium/phosphate due to concomitant kidney failure, etc.
Response 2:
As requested, we have better described the degenerative valve. This patient had severe mitral regurgitation related to mitral prolapse with fail leaflet.
The renal status of our patients has been checked beforehand. Only control patient had a sever kidney failure. We have clarified this in Table 1 of the new version submitted.
-Were tissues extracted from donors and processed for histology in fully-aseptic conditions? How was the possibility of environmental contamination excluded? Especially in light of the relatively-low degree of bacteria found in several vegetations…
Response 3:
Indeed, any environmental bacterial contamination was discarded by fully-aseptic conditions, from tissues extraction from donors to process for histology and electron microscopy. Tissue pieces were manipulated with clean sterile tools from extraction until observation. For electron microscopy for example, a clean and 100% ethanol-cleaned razor blade was used for cutting vegetations, in a ventilated biological safety cabinet. We would also like to highlight that within vegetations, bacteria were always found embedded in the tissue and not simply located on top of it, the later situation suggesting a contamination if having occurred. But in all conditions, bacteria were part of the tissue, even if in small number.
-Blood culture methodology used to confirm bacterial species found in IE valves is not described in the Materials and Methods. Were cultures taken from the valve leaflets themselves, from donor blood samples, etc?
Response 4:
Done as requested. The blood culture technique was used according to the standard model when establishing the diagnosis in patients. We have oriented towards a reference explaining this method. Please check the new submitted version, line: 130; page: 8.
-Manual quantitation of cellular organization, cell types, and their respective arrangements as imaged by SEM is basic and prone to bias. The hisological results presented in this manuscript are largely descriptive. With the widespread availability of automatic image analysis algorithms (inexpensive & open-source eg ImageJ/Fiji, or turnkey options from most microscope vendors), automatic quantitation of these parameters is desirable, as it enables quantifiable and blinded insight into these parameters.
Response 5:
Although the use of imageJ or Fiji would give relatively more objective results, but we think that it is more profitable with confocal or fluorescence microscopy where the cells are labeled. We recently published an article using a quantitative approach with the Fiji software on the analysis of bacteria platelet aggregates. For electron microscopy, we think that this software lacks sensitivity, on the one hand the samples are not marked, and therefore the cells do not have a clearly defined outline, but also, certain cells do not always have the same shape, like the example of platelets during activation. It is for these reasons that we opted for a semi-quantitative manual analysis, which takes a relatively long time by analyzing on average 100 image-fields per vegetation to arrive as exactly as possible at a conclusion for each cell type.
-Were investigators blinded to valve type/infection type during semi-quantitative analyses of disease components by histology?
Response 6:
For the classic histology part, these analyzes were carried out blind as long as they were part of the diagnosis. Regarding SEM / EDX analysis, although investigators were not blinded to valve type/infection during semi-quantitative analyses of vegetations cellular components, parallel semi-quantitative analysis were performed independently by two separate investigators and results of these analysis were compared.
-Why were multiple student’s t tests utilized for comparisons of EDX data? This type of multiple-comparisons testing requires ANOVA or similar.
Response 7:
We compared the samples two by two for each element, between the white and non-white areas. The multivariate ANOVA test followed by a post hoc Bonferroni test for example cannot be used in this case since the ANOVA requires the presence of 3 or more samples.
-Without further clinical patient data and n=1 biological replicates per condition, the purported presence of calcific deposits in veridans streptococci-infected valves is of little value. Calcium phosphate deposits are a common finding in diseased leaflets, and could easily be present due to underlying AI, AS, or other fibrocalcific valvular disease, kidney complications, etc.
Response 8:
In the interpretation of the results regarding the phosphocalcic deposits, we proceeded by the comparison of the viridans species (n = 2) versus the non-viridans species (n = 3). As you specified, the results exposed in this article cannot decide, but we suggest that it could be related to the species, especially since the literature is in favor, in particular by the case carried forward on S. salivarius as well as the implication of this group of species in calcification processes, other than in infective endocarditis.
- Rose R K. Binding characteristics of Streptococcus mutans for calcium and casein phosphopeptide. Caries Res. Sep-Oct 2000;34(5):427-31.doi: 10.1159/000016618.

Round 2
Reviewer 1 Report
The authors were quite responsive to my original concerns and have addressed them adequately.
Author Response
We thank you.
Reviewer 2 Report
Thank you for your thoughtful replies to my review, and for including a rich set of baseline data in Table 1. I appreciated the added methodological details.
Respectfully, and in full appreciation of the difficulties involved in studying this type of human disease, n=1 observational data cannot “reveal unique variations among infectious bacterial species”, nor can a novel methodological approach be appropriately validated with n=1 data per group.
Analysis of both aortic and mitral valves, as was disclosed to the other reviewer, presents an added source of variation between groups – native microstructures, surrounding microenvironment (blood pressure forces, hemodynamics, etc) of the aortic and mitral valves are notably different.
Author Response
Thank you for your thoughtful replies to my review, and for including a rich set of baseline data in Table 1. I appreciated the added methodological details.
Respectfully, and in full appreciation of the difficulties involved in studying this type of human disease, n=1 observational data cannot “reveal unique variations among infectious bacterial species”, nor can a novel methodological approach be appropriately validated with n=1 data per group.
We fully agree and recognize that n = 1 cannot confirm the significant difference for each bacterial species.
Indeed, the aim of our study was not to associate each species with a particular ultrastructure, something which requires a larger sample, and which possibly cannot be reached given the important variability between the patients. The key message of this article is to question the classic representation of vegetation: an infected and inflamed fibrin-platelet deposit, taking into consideration neither the bacteria involved nor the characteristics of the patient. Through our study, we have shown that the cellular composition of vegetations is far from being universal.
We have clarified this point in the new submitted version. Lines: 300-305; page: 19.
Analysis of both aortic and mitral valves, as was disclosed to the other reviewer, presents an added source of variation between groups – native microstructures, surrounding microenvironment (blood pressure forces, hemodynamics, etc) of the aortic and mitral valves are notably different.
We fully agree. We have clarified this point in the new submitted version. Lines: 307-311; pages: 19 and 20.